# Burden and Clinical Characteristics of Influenza and Its Complications in Children Across Multiple Epidemic Seasons

**DOI:** 10.3390/v17121574

**Published:** 2025-11-30

**Authors:** Arianna Dondi, Fiorentina Guida, Ludovica Trombetta, Maddalena De Peppo Cocco, Giulia Piccirilli, Laura Andreozzi, Eleonora Battelli, Pasquale Castaldo, Ilaria Corsini, Luca Pierantoni, Martina Franceschiello, Liliana Gabrielli, Monia Gennari, Dalila Periccioli, Tiziana Lazzarotto, Daniele Zama, Marcello Lanari

**Affiliations:** 1Pediatric Unit, IRCCS Azienda Ospedaliero-Universitaria of Bologna, 40138 Bologna, Italy; arianna.dondi@aosp.bo.it (A.D.); fiorentina.guida@unibo.it (F.G.); laura.andreozzi4@unibo.it (L.A.); eleonora.battelli@aosp.bo.it (E.B.); ilaria.corsini2@unibo.it (I.C.); luca.pierantoni@aosp.bo.it (L.P.); monia.gennari@aosp.bo.it (M.G.); daniele.zama2@unibo.it (D.Z.); marcello.lanari@unibo.it (M.L.); 2Department of Medical and Surgical Sciences, Alma Mater Studiorum, University of Bologna, 40137 Bologna, Italy; tiziana.lazzarotto@unibo.it; 3Speciality School of Paediatrics, Alma Mater Studiorum, University of Bologna, 40137 Bologna, Italy; pasquale.castaldo2@studio.unibo.it (P.C.); dalila.periccioli@studio.unibo.it (D.P.); 4Microbiology Unit, IRCCS Azienda Ospedaliero-Universitaria di Bologna, 40138 Bologna, Italy; giulia.piccirilli@aosp.bo.it (G.P.); martina.franceschiello@aosp.bo.it (M.F.); liliana.gabrielli@aosp.bo.it (L.G.)

**Keywords:** influenza virus, paediatric, children, complications

## Abstract

Seasonal influenza is a major cause of morbidity and hospitalization in children, with the potential for severe complications and considerable socioeconomic impact. We conducted a retrospective observational study including 1046 children aged 0–14 years with laboratory-confirmed influenza who accessed the Paediatric Emergency Department of a tertiary center in Bologna, Italy, across three consecutive epidemic seasons (2022–2025). While the entire cohort was analysed, particular attention was given to children with severe complications requiring hospitalization, for whom more detailed clinical and laboratory data were available. Overall, 12.3% of patients required hospitalization, and 6.1% experienced complications, most frequently influenza-associated encephalopathy, lower respiratory tract infections and myositis. Influenza A predominated overall (82.0%), except for in the last season, which saw a predominance of influenza B (57.4%), closely associated with myositis and elevated creatine phosphokinase levels. Younger age was consistently associated with increased severity and hospitalization. Intensive care admissions were rare (0.8%), and no deaths were recorded. Our findings suggest that, although influenza is generally self-limiting, younger children are at higher risk of complications. These results highlight the importance of active surveillance, careful monitoring of clinical manifestations and targeted paediatric vaccination strategies to reduce the burden of seasonal influenza.

## 1. Introduction

Seasonal influenza epidemics, mainly caused by influenza A and influenza B viruses, pose a major health threat to children, leading to significant morbidity, hospitalizations, and mortality, particularly in those under five years of age or with an underlying medical condition [1]. Seasonal influenza viruses are primarily transmitted via respiratory droplets and short-range aerosols, with an incubation period ranging from 1 to 4 days [2]. The onset of illness is typically abrupt, with nonspecific systemic manifestations such as fever, chills, headache, myalgia, and malaise, frequently accompanied by respiratory symptoms. In children, however, influenza may present solely with febrile seizures, even in the absence of other significant systemic signs or respiratory involvement [3,4,5,6].

Although influenza virus infection is typically self-limiting within 5 to 10 days, it may lead to severe and potentially life-threatening complications [3]. Notably, evidence indicates that influenza A and B infections exhibit comparable clinical severity in pediatric populations [7]. These complications, which may vary according to viral type, are heterogeneous and are generally classified as respiratory or non-respiratory.

Respiratory complications include both upper respiratory tract diseases—such as otitis media, parotitis, sinusitis, and laryngotracheobronchitis—and lower respiratory tract infections (LRTIs), including bronchiolitis, bronchitis, reactive airway disease, pneumonia, respiratory failure, and acute respiratory distress syndrome (ARDS). Influenza virus infection can also exacerbate pre-existing chronic respiratory conditions such as asthma and cystic fibrosis [5].

Non-respiratory complications are various, involving multiple organ systems. Cardiovascular involvement, particularly myocarditis and pericarditis, has been more frequently associated with influenza A [8]. Conversely, musculoskeletal complications are predominantly linked to influenza B infection, with acute myositis representing the most common findings. This condition occurs more frequently in males and typically presents with myalgia localized to the lower limbs—especially the calves—often accompanied by refusal to walk or gait disturbances. Laboratory abnormalities commonly include elevated serum creatine phosphokinase (CPK) and transaminase levels [9].

Neurological complications, grouped under the term influenza-associated encephalopathy or encephalitis (IAE), usually follow an initial respiratory infection and seem to reflect a dysregulated host inflammatory response. They are most often observed in hospitalized patients and include febrile and afebrile seizures, acute necrotizing encephalopathy, aseptic meningitis, myelitis, secondary bacterial meningitis, brain abscess, and, more rarely, cerebral infarction or Reye syndrome [10].

Globally, the percentage of children with influenza who require hospitalization generally ranges between 1% and 5%, with higher rates among children under 5 years of age and in low-income countries [11]. In Italy, the most recent data indicate that the percentage of children with influenza who are hospitalized is roughly the same [12,13].

Antiviral medications can shorten the duration of influenza illness [3,14], with greatest efficacy when administered soon after symptom onset [15]. Early treatment, ideally within 48 h of symptom onset, is recommended for children who are hospitalized, have severe or progressive disease, or possess underlying conditions that increase the risk of complications, in which case treatment should be offered regardless of illness duration [5,16].

In Italy, the prevalence of influenza in children aged 0 to 14 in 10 seasons from 2010 to 2020 ranged from 3.4% in 2013–2014 to 8.6% in 2010–2011 [12], with influenza B virus prevailing from March 2023 onwards [17]. Nevertheless, the incidence of each seasonal epidemic period may vary depending on multiple factors. The circulation of other infectious diseases can influence the number of positive registered cases. Notably, public health measures adopted to contain the COVID-19 pandemic significantly reduced influenza transmission. For instance, influenza cases in March 2022 declined sharply, and the 2020–2021 season recorded the lowest global incidence ever documented [3]. Although influenza activity remained relatively low during the 2021–2022 season, the subsequent seasons (2022–2023 and 2023–2024) were characterized by a marked resurgence in incidence. This trend coincided with declining vaccination coverage and was accompanied by surveillance data indicating increased disease severity, reflected in higher hospitalization and mortality rates [16,18].

Although influenza in children has been extensively studied, significant gaps remain in the understanding of cases requiring hospitalization. In children, large-scale epidemiologic cohort studies are limited, and much of the existing literature focuses on the occurrence of complications rather than on the clinical characteristics and risk factors that may predispose to severe disease. In addition, evidence on the management and outcomes of influenza in the paediatric emergency department (PED), as well as the treatment of hospitalized cases, including the timing and use of antiviral therapy and the management of complications, remains incompletely characterized.

Aim of the study: We conducted a monocentric retrospective observational study of laboratory-confirmed influenza cases in children aged 0–14 years referred to our center over three epidemic seasons. We aimed to characterize clinical features and management of influenza, as well as the spectrum and burden of severe influenza-associated complications, in children presenting to the PED and those requiring hospitalization.

## 2. Materials and Methods

### 2.1. Study Design and Population

Across three consecutive influenza epidemic seasons, from 1 October 2022 to 30 April 2025, we retrospectively enrolled children aged 0 to 14 years who accessed the PED of the IRCCS Azienda Ospedaliero-Universitaria of Bologna and tested positive for influenza A or B at admission (only a single respiratory sample per patient at diagnosis was considered). Ethical approval was obtained from the local Ethics Committee (Protocol No. 657/2023/Oss/AOUBo).

### 2.2. Microbiology

At admission, a nasopharyngeal swab was collected for each patient and analysed by molecular method using Allplex™ RV Master Assay (Seegene Inc., Seoul, Republic of Korea), a multiplex real-time reverse transcription polymerase chain reaction (RT-PCR) system that enables simultaneous detection of Influenza A virus (Flu A), Influenza B virus (Flu B), Human metapneumovirus (MPV), SARS-CoV-2, parainfluenza virus (PIV), Human adenovirus (AdV), rhinovirus and respiratory syncytial virus (RSV). Specifically, RNA was extracted using the Seegene STARMag 96 × 4 Universal Cartridge Kit on the Seegene STARlet system (Seoul, South Korea) starting from 300 microliters of samples eluted in 100 microliters. Five microliters was used for amplification according to the manufacturer’s instructions using the CFX96 Real-Time Detection System (Bio-Rad, Hercules, CA, USA). The limit of detection is equal to 50 copies/reaction for SARS-CoV-2 and 100 copies/reaction for all the remaining targets. Samples with Ct value ≤ 40 or 41 were considered positive for MPV, Flu B, HRV, RSV and SARS-CoV-2, PIV, AdV, Flu A, respectively.

### 2.3. Data Collection

For each patient, demographic data, the influenza virus type (A or B) and clinical data were collected.

For children requiring hospitalization, detailed information was collected on the characteristics and duration of symptoms, including fever, respiratory, gastrointestinal, and systemic manifestations. The presence of underlying conditions was also recorded, encompassing oncologic/immunosuppressive, neurologic/neuromuscular, respiratory, renal/urologic, cardiovascular, endocrinologic, gastroenterological/hepatic, genetic/metabolic, hematologic conditions, obesity, or other complex chronic diseases.

Laboratory data included white blood cell count with differential, inflammatory markers, liver transaminases, renal function parameters, and CPK levels. Clinical outcomes were also documented, including length of hospital stay (LOS), the need for paediatric intensive care unit (ICU) admission, and the occurrence of influenza-related complications, such as myositis, myocarditis, encephalitis, lower respiratory tract infections (LRTIs), respiratory failure, head and neck infections, or bacteremia.

Additional data on therapeutic management were collected, including antiviral and antibiotic therapy, oxygen supplementation, high-flow support, and mechanical ventilation.

### 2.4. Statistical Analysis

Statistical analysis was performed with RStudio 4.4.1 (1 June 2025). Categorical variables were summarized as absolute frequencies and percentages. Continuous variables were described using median and interquartile range (IQR).

Associations between dichotomous variables were assessed using Fisher’s exact test. Adjusted odds ratios (aORs) for age were estimated through logistic regression models. Logistic regression was also employed for multivariable post hoc analysis. For comparisons involving continuous variables, the Kruskal–Wallis test was used. When appropriate, pairwise post hoc comparisons were conducted using Dunn’s test with correction for multiple testing. A two-sided *p* value < 0.05 was considered statistically significant.

## 3. Results

### 3.1. Analysis by Epidemic Seasons

Among the observational period, 1046 children tested positive for influenza: in season 1 (2022–2023) 526 patients (50.3%), in season 2 (2023–2024) 372 patients (35.6%) and in season 3 (2024–2025) 148 patients (14.5%). The main data on the number of influenza-positive patients relative to the total PED visits across the three epidemic seasons are summarized in Table 1.

The prevalence of specific influenza types is shown in Figure 1a. Overall, 12.3% of patients (129/1046) required hospitalization (Figure 1b) and 6.1% (64/1046) experienced at least one influenza-related complication (Figure 1c).

### 3.2. Analysis by Hospitalization Rate

A focused analysis on hospitalized patients revealed that the median age was significantly lower compared with children managed as outpatients (*p* < 0.001). Additionally, children admitted to the paediatric department with influenza A were significantly younger than those with influenza B (*p* < 0.001). No significant differences were observed in sex distribution. LOS was significantly longer in patients who had complications (*p* < 0.05) or comorbidities (*p* < 0.01); by contrast, patients younger than 24 months of age were not hospitalized for a significantly longer period than other patients. Influenza A represented the predominant type among hospitalized patients. Specifically, 99 out of 858 influenza A cases (11.5%) and 30 out of 188 influenza B cases (16.0%) required hospitalization. The overall distribution of influenza types differ significantly between hospitalized and non-hospitalized children in the age-adjusted analysis. The main findings are summarized in Table 2.

Regarding clinical presentation, patients with influenza A and B exhibited comparable symptom profiles (Table 3). Fever was the most prevalent manifestation, with a similar duration observed in both viral types, followed by upper respiratory tract symptoms. Myalgias were reported more frequently in influenza B cases (*p* < 0.001), whereas gastrointestinal symptoms were slightly more common in influenza A, although this difference did not reach statistical significance.

Complications occurred significantly more frequently in children with influenza B (*p* < 0.05) compared with influenza A. Respiratory complications, mainly LRTIs, were observed more frequently in patients with influenza A, who also exhibited a higher incidence of IAE. Conversely, myositis and rhabdomyolysis were reported more often in patients with influenza B. Other non-respiratory complications, including bacteremia (n = 2, 1.6%), myocarditis (n = 1, 0.8%), and head or neck infections (n = 1, 0.8%), were rare and occurred at comparable rates in both influenza types.

Laboratory findings revealed some distinct patterns. Children with influenza A showed significantly higher white blood cell count—although according to the age-adjusted analysis this shows only a trend toward statistical significance (*p* = 0.07)—and neutrophil count. CPK levels were markedly elevated in influenza B, consistent with the higher rate of myositis. The main results are summarized in Table 3.

### 3.3. Antiviral and Antibiotic Therapy

In our study, antiviral therapy with Oseltamivir was administered in 15 patients, more frequently during season 2 (*p* < 0.01) and especially in patients admitted to ICU (*p* < 0.001). Oseltamivir was prescribed for a duration of 5 to 10 days at pediatric dosage, and 86.7% (n = 13) of patients who received antiviral therapy were also administered antibiotics. Among hospitalized patients, 33.3% (n = 43) underwent antibiotic treatment, and beta-lactams was the most frequently prescribed class of antibiotics (n = 42).

### 3.4. Analysis by Complications

In the results of our analysis, statistically significant differences emerged between patients with and without complications. In particular, patients who developed complications had a lower median age (median age 53.9 vs. 30.4 months; IQR 59.9 vs. 46.3; *p* < 0.001). Patients with influenza B infection were at higher risk of complications (aOR 2.43, *p* < 0.001; 95% CI 1.34–4.28). Regarding laboratory tests, white blood cells (median 11,145/mm^3^ vs. 8345/mm^3^, IQR 6852/mm^3^ vs. 6425/mm^3^; *p* < 0.05), particularly neutrophils (median 6750/mm^3^ vs. 5045/mm^3^, IQR 5982/mm^3^ vs. 4812/mm^3^; *p* < 0.05), were elevated in patients with complications, while no differences were observed in CRP values (*p* = 0.66). In the multivariate analysis, younger age (aOR 0.98, CI 0.96–0.99, *p* < 0.01) and CPK levels (aOR 1.001, CI 0.0001–1.004, *p* < 0.05) were identified as independent risk factors for identifying patients at risk of complications.

As for the individual complications, no significant difference was observed in sex distribution between patients with and without myositis. However, children developing myositis were significantly older compared to those without (median age 87.4 vs. 51.9 months; IQR 36.0 vs. 60.2; *p* < 0.05). Myositis was strongly associated with influenza B infection (*p* < 0.001; OR 13.22, 95% CI 3.86–57.59). The analysis also showed that having myositis increases the risk of hospitalization (*p* < 0.001; OR 21.15, 95% CI 6.14–92.55) with myalgia being the symptom most closely associated with myositis (*p* < 0.001; OR 443.3, 95% CI 40.19–16,384). Patients with myositis had significantly higher levels of aspartate transaminase (AST), alanine aminotransferase (ALT), creatinine and CPK. In the multivariate analysis, CPK values are an independent risk factor for myositis (*p* < 0.05, OR 1, 95%, CI 1.002–1.010).

IAE occurred more frequently in younger patients (median age 25.4 vs. 53.1 months; IQR 20.0 vs. 60.5; *p* < 0.001). Additionally, children with IAE presented higher white blood cell counts (12,690 vs. 8740/µL, *p* < 0.05), particularly neutrophils (7345 vs. 5150/µL, *p* < 0.01). In patients with IEA, antibiotics were used more frequently (*p* < 0.01, OR 0.15 CI 0.016–0.681), while no differences were observed in the use of antivirals. In the multivariate analysis, a younger age at admission (*p* < 0.01, OR 0.954, 95% CI 0.919–0.992) was an independent risk factor for IAE.

Patients with LRTIs were younger than those without this complication (median age 15.3 vs. 53.4 months; *p* < 0.01, IQR 12.3 vs. 59.9) and presented higher white blood cell counts (12,235 vs. 8860/µL, *p* < 0.05), particularly neutrophils (7990 vs. 5265/µL, *p* < 0.05). This subset of patients exhibited greater need for pharmacological interventions, including antiviral treatment (*p* < 0.01) and antibiotic administration (*p* < 0.001). In the multivariable analysis, younger age (*p* < 0.05, OR 0.97; 95% CI 0.94–0.99) was identified as an independent predictor of the risk of LRTIs in the influenza patient population.

Younger patients, defined in our study as those under 24 months of age, exhibited a more severe clinical course, with a higher incidence of complications (*p* < 0.001, OR 0.36; 95% CI 0.21–0.63). This subgroup of patients was infected more frequently with type A (*p* < 0.05, OR 1.54; 95% CI 1.01–2.39), with a lower incidence of myositis but a higher risk of LRTIs. In this age group, higher COVIs of hospital admission (*p* < 0.001, OR 0.34; 95% CI 0.23–0.51), oxygen therapy (*p* < 0.001, OR 0.11; 95% CI 0.036–0.320), and antibiotic use (*p* < 0.05, OR 0.42; 95% CI 0.18–0.92) were observed, but not a higher risk of ICU admission. No difference in the LOS was observed between the younger and older patient groups. In the younger group, the presence of comorbidities did not result in longer hospital stay.

## 4. Discussion

This study offers a comprehensive and analytical perspective on the disease distribution of influenza among children presenting to a tertiary university PED, showing an overall incidence of hospitalization and complications of 12.3% and 6.1% of cases, respectively, across three consecutive epidemic seasons. The overall median age was 52.2 months with a statistically significant difference across seasons, while the gender distribution remained balanced, with no significant seasonal variation.

Although our findings align with the generally self-limiting course of influenza in children, specific subgroups such as younger children are at increased risk of developing complications or requiring hospitalization. As widely reported in the literature, the subgroup of younger patients exhibited a greater incidence of complications and hospitalizations, with more severe clinical courses characterized by increased need for oxygen therapy and antibiotic use. This increased vulnerability may be closely linked to age-dependent differences in innate immune responses. In early childhood, the relative immaturity of pathogen-recognition pathways results in an altered cytokine profile during influenza infection, with reduced production of IL-12 and IFN-γ—key mediators of antiviral defence—and, conversely, higher levels of pro-inflammatory cytokines such as IL-6 and IL-1β, as well as the anti-inflammatory cytokine IL-10. Such an imbalance may simultaneously impair viral clearance and promote an exaggerated inflammatory response, a combination that has been associated with more severe disease in pediatric patients. Moreover, emerging evidence suggests that, in children, severe influenza infection may be driven less by uncontrolled viral replication and more by a dysregulated host inflammatory response, which can subsequently shift toward a state of relative immunosuppression. This post-inflammatory phase may increase susceptibility to secondary bacterial infections, further contributing to clinical deterioration [19,20].

As well demonstrated, influenza virus activity markedly decreased following the COVID-19 pandemic, and during the 2020–2021 season, it was the lowest ever recorded worldwide [21]; this phenomenon aligns with previous reports and is likely attributable to global public health measures implemented to counteract SARS-CoV-2 transmission, including mask-wearing, social distancing, school closures, teleworking, and reduced travel [3]. These non-pharmacological interventions (NPIs) also altered the typical seasonality of influenza, leading to the disappearance of the usual seasonal peaks and, in some cases, to the emergence of out-of-season epidemics following the lifting of restrictions. Another relevant aspect is the phenomenon of ‘immune debt’: the reduced circulation of the influenza virus during the pandemic led to a decline in population immunity, increasing susceptibility and facilitating the occurrence of potentially more intense influenza waves once the NPIs were lifted [22,23,24]. Indeed, in our study, seasons 1 and 2 were characterized by higher influenza cases, in line with trends reported in other studies, and were accompanied by declining reported vaccination coverage and greater disease severity, as evidenced by higher rates of hospitalization and complications [12,25,26]. Consistently, our data showed a peak of hospitalizations and complications in season 1. In the following seasons 2 and 3, influenza cases declined, a trend that may be related to the gradual increase in influenza vaccination coverage among children in our region. The Italian Department of Prevention, Research and Health Emergencies recommends annual influenza vaccination of children aged 6 months to 6 years, as well as high risk and medically vulnerable patients and their caregivers or household contacts [27]. In our region Emilia-Romagna, influenza vaccination coverage among children aged 0–14 years has remained low but with a slight increase over the past three seasons, from approximately 3.0% in season 1 up to a maximum coverage of 19.9% in season 3 [28,29,30,31].

It is important to note that our study was not designed to estimate influenza cases in the general pediatric population, as only symptomatic children attending the PED were included. Moreover, the Italian healthcare system ensures that every child has an assigned primary care pediatrician, which may substantially influence healthcare-seeking behaviours and consequently result in different case distributions compared with settings where healthcare structures are organized differently.

The seasonal distribution of viral types observed in our study is consistent with previous reports, which generally show a predominance of influenza A (82.0%, n = 858). However, notable interannual fluctuations were observed, with certain seasons exhibiting a higher prevalence of influenza B [12,17], as observed in our cohort during season 3.

Influenza A virus was found to affect younger children more frequently. Consistent with this and with the seasonal pattern of viral types, when the age-adjusted analysis was performed, a shift in the trend toward non-statistical significance was observed. This supports the notion that influenza type is not uniformly stratified across age groups, and therefore seasons in which one influenza type predominated over the other are characterized by different age distributions. Influenza A was significantly associated with respiratory and neurological complications, as well as neutrophilic leukocytosis. This latter finding has been reported in a recent study comparing the clinical features of influenza A and B, although no statistical significance was observed in that analysis [32].

Conversely, influenza B was associated with a higher rate of complications, particularly myositis, which is characterized by myalgia and elevated AST, ALT, creatinine and CPK levels. The greater burden of influenza B also corresponded with the highest incidence of complications and a high hospitalization rate observed during season 3, when this strain predominated.

In our study the clinical severity of influenza A and B was comparable, consistent with current evidence reporting similar outcomes between the two types of influenza and sometimes even higher severity for influenza A [7,33]. In our cohort, influenza B infections were observed to be more frequently associated with complications than those caused by influenza A, particularly when the analysis was adjusted for age. In addition, after removing the age-related bias, influenza type emerged as a predictor of hospitalization—with influenza B carrying a higher risk. Moreover, this analysis allows us to conclude that the strikingly different distribution of viral types across seasons is almost entirely explained by age rather than by intrinsic viral factors. In essence, age is not a simple covariate but a true epidemiological driver that shapes the entire dynamics of pediatric influenza.

Our data revealed that 12.3% (n = 129) of patients presenting to the PED were hospitalized, with a significant difference between seasons (Figure 1c); this percentage is lower than that reported in recent studies, which show that approximately 20–23% of pediatric patients with confirmed influenza evaluated in emergency departments or hospital settings require hospitalization [34,35]. The lower hospitalization rate observed in our cohort may reflect the well-developed community care network around our center supporting efficient outpatient management. In addition, a different population of patients, seasonal variations in circulating influenza strains and local differences in admission thresholds or practice patterns may have differed from that of previous reports. Moreover, oxygen supplementation (n = 21, 2.0%, *p* = 0.08), non-invasive ventilation (n = 7, 0.7%, *p* = 0.06), and ICU admission (n = 8, 0.8%, *p* = 0.88) remained rare events throughout the study period, without clearly significant variation across seasons. Most children (96.3%) were healthy without previous comorbidities, and no deaths were recorded.

In line with expectations, hospitalized patients were significantly younger than outpatients (*p* < 0.001). The mean length of hospital stay was approximately 4.5 days, with no significant differences across age groups. Interestingly, hospitalization was notably longer among patients with underlying comorbidities or those who developed complications.

The complications analysed in our study that required hospitalization occurred in 5.7% of patients (n = 60), a lower percentage than in the data found in the literature [36]. This lower rate may be explained by the retrospective design of the study and the fact that mild (e.g., some myositis or upper respiratory complications) complications were not included. Complications were less frequent in season 2, whereas the burden was considerably higher in seasons 1 and 3.

Overall, the pattern of complications observed in our cohort is broadly consistent with what has been reported in contemporary studies, although some differences in relative frequencies emerged. Respiratory complications among hospitalized children with influenza occurred in 14.7% of patients, slightly lower than those reported in the literature (pneumonia 17–28% of cases, bronchiolitis 0.5–5% of cases and acute respiratory failure 5–5.3% of cases). In contrast, neurological complications, which usually represent 8–11% of hospitalized cases, were comparatively more frequent in our cohort (n = 23, 17.8%) and myositis, generally reported in 7–10% of hospitalized children, was relatively common in our cohort (n = 15, 11.6%), with a predominance in season 3. Other non-respiratory complications such as bacteremia, myocarditis and head/neck infections remained rare (<2%), consistent with published data (0.9–2%). It could be hypothesized that this different spectrum of complications is partly influenced both by the characteristics of the circulating influenza viruses during the study period and the specific criteria used to classify complications in our retrospective analysis [16,18,37,38,39,40].

The clinical profile of complications also differed: respiratory complications we described, were 19 cases of LTRIs which predominated in season 1. Between the non-respiratory complications, IAE (n = 23) were the most frequent, while myositis, also common (n = 15), was predominant in season 3. Other non-respiratory complications such as bacteremia, myocarditis and head/neck infections were extremely rare. It could be hypothesized that this spectrum of complications is partly influenced by the characteristics of the circulating influenza viruses during the study period.

Although neurological complications are more common in children with chronic neurological conditions and are associated with worse outcomes [10,41,42], the limited number of patients with comorbidities in our study prevented us from determining whether specific types of complications correlate with comorbidities or whether comorbidity type could predict different outcomes. IAE is among the most reported neurological complications of influenza infections [10]. In our study, younger age is an independent risk factor for developing neurological complications, consistent with previous findings that age ≤ 5 years is associated with increased IAE risk [41]. Nearly all cases in our cohort presented with febrile seizures, with only two exceptions: one case of encephalitis and one case of Posterior Reversible Encephalopathy Syndrome (PRES), both linked to influenza A.

Influenza is known also to be a common cause of LRTIs in young children [1]. In our study we described 19 cases of LRTIs, which predominated in season 1, including 12 cases of acute respiratory failure (5 of whom were admitted to ICU), 4 cases of acute bronchiolitis and 3 cases of bacterial pneumonia, one of which was complicated by pleural empyema. In our cohort, patients who developed LRTIs were significantly younger and with a longer duration of illness. Also, this is the only subgroup of patients that significantly needed more supportive care (including more frequent use of oxygen) and pharmacological interventions, both antiviral and antibiotic treatments.

Myositis is the most common musculoskeletal complication of influenza, predominantly influenza B, and usually presents itself with myalgia in the lower limbs in older and male patients [9]. Indeed, in our study, patients who developed myositis were older and often reported myalgia, consistent with the literature, but we did not find a significant difference in gender distribution. Only elevated CPK proved to be an independent risk factor for the development of myositis. The association between influenza B and myositis is likely mediated by the virus’s pronounced muscle tropism, which consequently leads to elevated serum AST and CPK levels as a result of muscle injury.

Usually, influenza disease has a self-limiting course regardless of treatment with antiviral medications [3]. The clinical benefit of antiviral treatment of influenza is greatest when started soon after illness onset, both in outpatient and in hospitalized patients with high-risk medical conditions [15]. Meta-analyses of randomized controlled trials reported that Oseltamivir treatment of uncomplicated influenza significantly reduced illness duration and risk of otitis media, if started within 2 days of symptom onset [14]. In our study, antiviral therapy was used more in seasons 2 and 3 and especially in patients admitted to ICU. Oseltamivir was the only antiviral drug prescribed in our center, according to paediatric dosing guidelines for 5 to 10 days. Antiviral therapy was prolonged for more than 5 days mainly in patients with a documented or suspected immunocompromising condition or severe lower respiratory tract disease, such as pneumonia or acute respiratory distress syndrome (ARDS), according to the 2019 update of the guidelines published by the Infectious Diseases Society of America [43]. Current recommendations suggest that empirical antibiotic therapy for suspected bacterial coinfection should be added to antiviral treatment in cases of severe disease and should be considered in patients who show no improvement after 3–5 days of antiviral therapy [43]. Aligning with this, in our cohort, most patients who received antiviral therapy had either previously received or were concurrently receiving antibiotic treatment. This can be explained by the fact that hospitalized patients often present with severe complications, prompting the initiation of empiric antibiotic therapy, which is then discontinued if microbiological tests are negative.

The strengths of this study include its substantial sample size (n = 1046) and the routine use of molecular diagnostics, which enhance the accuracy of detecting viral infections.

Despite its methodological strengths, this study presents several limitations, primarily related to its single-center retrospective design. First, a potential selection bias cannot be excluded, thereby possibly overlooking milder cases within the broader population. Second, we acknowledge the possibility of selection bias in patient enrollment from the Emergency Department, as the absence of standardized criteria for virological testing meant that the decision to test was left to the attending physician’s clinical judgment. This likely resulted in preferential testing of younger children, patients with comorbidities, or those considered for hospitalization, potentially leading to an overrepresentation of clinically complex cases in our sample—a recognized study limitation. Third, incomplete data on individual vaccination status and the timing of antiviral administration hinder a comprehensive evaluation of the efficacy of vaccination and the impact of antiviral therapy on clinical outcomes. In addition, the generalizability of our findings may be limited to centers employing analogous diagnostic tests and populations with comparable demographic and epidemiological characteristics. Finally, another limitation of our study is that we did not have access to quantitative viral load data for all patients. Viral detection was performed using a standard molecular swab, and only a subset of patients underwent a preliminary rapid antigen test, which provided an indirect estimate of viral load. Having quantitative viral load measurements would indeed have strengthened our analysis, as it would have allowed us to verify within our cohort what several studies have suggested—namely, that viral load tends to be higher in children and in hospitalized patients, and that treatment with oseltamivir initiated within the first 48 h of symptom onset is associated with a faster viral clearance time [44]. This aspect will be important to address in future studies.

## 5. Conclusions

By its nature, influenza occurs within a limited temporal window, concentrating cases over a short period and posing a substantial clinical and organizational challenge for emergency departments. In recent years, the real burden of influenza has been increasingly appreciated, encompassing the medical and socioeconomic impacts that childhood seasonal influenza can impose on hospitalization rates, outpatient visits, medications, school absences, and missed workdays for caregivers. Moreover, the persistently low influenza vaccination coverage in children underscores the urgent need for targeted immunization strategies to reduce the incidence of severe disease, prevent complications, and alleviate the socioeconomic burden associated with seasonal influenza. This study provides important insights into the clinical course and burden of influenza in children across three consecutive epidemic seasons. Our findings indicate that, although influenza is generally self-limiting in most pediatric cases, specific subgroups—such as younger children—are at increased risk of complications or hospitalization. Our results emphasize the importance of active surveillance of high-risk subgroups and highlight the need to carefully monitor certain signs and symptoms for their strong association with specific complications, particularly muscular signs in influenza B infections, which are consistently associated with myositis.

## Figures and Tables

**Figure 1 viruses-17-01574-f001:**
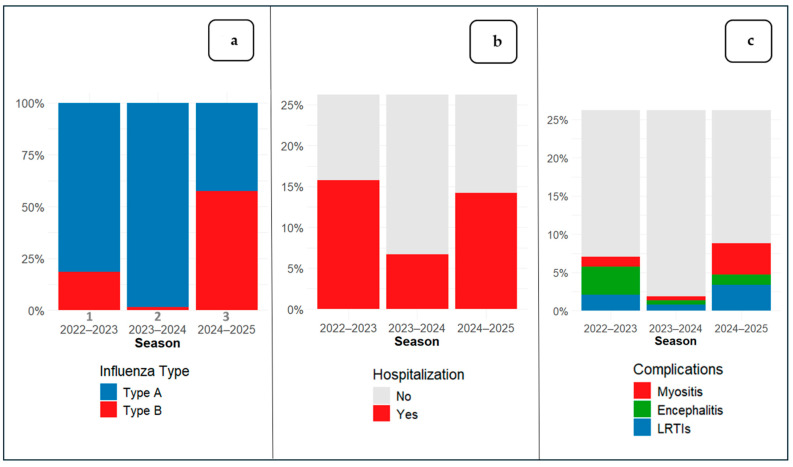
Percentage distribution of influenza types (**a**), hospitalizations (**b**) and complications (**c**) by seasons; (LRTIs = lower respiratory tract infections).

**Table 1 viruses-17-01574-t001:** Analysis by epidemic seasons (aOR = adjusted Odds Ratio; IQR = interquartile range; M = male; F = female).

Epidemic Season	Season 1 2022–2023	Season 2 2023–2024	Season 3 2024–2025	Total	*p*-Value	aOR (95%, CI)	*p*-Value (Adjusted)
**Flu-positive, n (%)**	526	372	148	1046			
**Age (months ± IQR)**	59.2 ± 65	48 ± 52.7	49.4 ± 63.7	52.2 ± 60.7	<0.001	0.99 (0.99–0.996)	<0.01
**Gender**							
**M, n (%)**	299 (56.8)	202 (54.3)	71 (48.0)	572 (54.7)	0.16	0.85 (0.66–1.08)	0.18
**F, n (%)**	227 (43.2)	170 (45.7)	77 (52.0)	474 (45.3)			
**Type of flu**							
**A, n (%)**	429 (81.5)	366 (98.4)	63 (42.6)	858 (82.0)	<0.001	1.001 (0.73–1.38)	0.99
**B, n (%)**	97 (18.5)	6 (1.6)	85 (57.4)	188 (18.0)			
**Complications**							
**Yes, n (%)**	41 (7.8)	9 (2.4)	14 (9.5)	64 (6.1)	<0.01	0.47 (0.27–0.8)	<0.01
**No, n (%)**	485 (92.2)	363 (97.6)	134 (91.5)	982 (93.9)			
**Comorbidity**							
**Yes, n (%)**	25 (4.8)	6 (1.6)	8 (5.4)	39 (3.7)	<0.05	0.54 (0.27–1.05)	0.07
**No, n (%)**	501 (95.2)	366 (98.4)	140 (94.6)	1007 (96.3)			
**Hospitalization**							
**Yes, n (%)**	83 (15.8)	25 (6.7)	21 (14.2)	129 (12.3)	<0.001	0.46 (0.31–0.67)	<0.001
**No, n (%)**	443 (84.2)	347 (93.3)	127 (85.8)	917 (87.7)			

**Table 2 viruses-17-01574-t002:** Characteristics of patients who required hospitalization (aOR = adjusted Odds Ratio; IQR = interquartile range; M = male; F = female).

Characteristic	Not Hospitalized	Hospitalized	Total	OR (95% CI)	*p*-Value	aOR (95% CI)	*p*-Value (Adjusted)
**Age (months ± IQR)**	54.8 ± 58.1	34.4 ± 56.4	52.2 ± 60.7		<0.001		
**Gender**							
**M, n (%)**	497 (54.2)	75 (58.1)	572 (54.7)	1.17 (0.79–1.73)	0.45	1.19 (0.82–1.74)	0.36
**F, n (%)**	420 (45.8)	54 (41.9)	474 (45.3)				
**Type of flu**							<0.05
**A, n (%)**	759 (82.8)	99 (76.7)	858 (82.0)	1.45 (0.90–2.29)	0.11	1.64 (1.03–2.55)	
**B, n (%)**	158 (17.2)	30 (23.3)	188 (12.0)				
**Epidemic season**					<0.001		<0.001
**season 1, n (%)**	443 (84.2)	83 (15.8)	526	reference	reference	reference	
**season 2, n (%)**	347 (93.3)	25 (6.7)	372	0.38 (0.23–0.60)	<0.05	0.34 (0.21–0.54)	<0.05
**season 3, n (%)**	127 (85.8)	21 (14.2)	148	0.88 (0.51–1.46)	0.64	0.80 (0.46–1.33)	0.4
**All, n (%)**	917 (87.7)	129 (12.3)	1046 (100)				

**Table 3 viruses-17-01574-t003:** Analysis by influenza type (aOR = adjusted Odds Ratio; IQR = interquartile range; M = male; F = female; IAE = influenza-associated encephalopathy or encephalitis; LRTIs = lower respiratory tract infections; WBC = white blood cells; CRP = C-reactive protein; AST = aspartate transaminase; ALT = alanine aminotransferase; CPK = creatine phosphokinase).

	Flu A	Flu B	Total	OR (95% CI)	*p*-Value	aOR (95% CI)	*p*-Value (Adjusted)
**Age (months ± IQR)**	50.3 ± 58.2	64.4 ± 61.2	52.2 ± 60.7		<0.001		
**Gender**							
**M, n (%)**	480 (56.0)	92 (49.0)	572 (55.0)	0.75 (0.54–1.05)	0.09	0.74 (1.002–1.010)	0.07
**F, n (%)**	378 (44.0)	96 (51.0)	474 (45.0)				
**Epidemic season**					<0.001		
**season 1, n (%)**	429 (50.0)	97 (51.6)	526 (50.3)	reference	reference	reference	reference
**season 2, n (%)**	366 (42.7)	6 (3.2)	372 (35.6)	0.07 (0.03–0.15)	<0.001	0.08 (0.03–0.16)	<0.001
**season 3, n (%)**	63 (7.3)	85 (45.2)	148 (14.1)	5.97 (4.04–8.88)	<0.001	6.41 (4.30–9.63)	<0.001
**Symptoms**							
**Fever, n (%)**	94 (11.0)	30 (16.0)	124 (11.9)	0.96 (0.16–10.2)	1	0.99 (0.21–7.10)	0.99
**Days of fever, (days ± IQR)**	4 ± 4.25	4 ± 3			0.26	0.83 (0.65–1.02)	0.11
**Upper airway inflammation, n (%)**	74 (8.6)	20 (10.6)	94 (9.0)	0.59 (0.23–1.51)	0.26	0.65 (0.27–1.57)	0.32
**Myalgias, n (%)**	4 (0.5)	8 (4.3)	12 (1.1)	7.82 (1.91–38.6)	<0.001	7.72 (2.03–34.39)	<0.01
**Headache, n (%)**	5 (0.6)	1 (0.5)	6 (0.6)	0.61 (0.01–5.8)	1	0.48 (0.02–3.27)	0.52
**Gastrointestinal, n (%)**	40 (4.7)	7 (3.7)	48 (4.6)	0.42 (0.14–1.13)	0.09	0.36 (0.13–0.91)	<0.05
**Complications**							
**Yes, n (%)**	45 (5.2)	19 (10.1)	64 (6.1)	2.03 (1.09–3.65)	<0.05	2.34 (1.29–4.08)	<0.01
**No, n (%)**	813 (94.8)	169 (89.9)	982 (93.9)				
**Complication Type**							
**Myositis, n (%)**	4 (0.5)	11 (5.9)	15 (1.4)				
**IAE, n (%)**	20 (2.3)	3 (1.6)	23 (2.2)				
**LRTIs, n (%)**	16 (1.9)	3 (1.6)	19 (1.8)				
**Comorbidity**							
**Yes, n (%)**	31 (3.6)	8 (4.3)	39 (3.7)	1.18 (0.46–2.69)	0.67	1.20 (0.51–2.55)	0.65
**No, n (%)**	827 (96.4)	180 (95.7)	1007 (96.3)				
**Laboratory exams**							
**WBC (n/mmc ± IQR)**	10,020 ± 5780	7530 ± 8070	9255 ± 6502		<0.01		0.07
**Neutrophils (n/mmc ± IQR)**	6260 ± 5090	3760 ± 5540	5588 ± 5562		<0.01		<0.05
**Lymphocytes (n/mmc ± IQR)**	2380 ± 2520	2149 ± 2260	2360 ± 2320		0.54		0.72
**CRP (mg/dL ± IQR)**	1.05 ± 3.28	0.64 ± 2.49	1.01 ± 3.22		0.31		0.64
**AST (U/L ± IQR)**	50 ± 24	58.5 ± 48.8	51 ± 28		0.14		0.15
**ALT (U/L ± IQR)**	20 ± 15.8	26.5 ± 21.5	21.5 ± 16		0.06		0.86
**CPK (U/L ± IQR)**	94 ± 66	218 ± 3718	106 ± 131		<0.01		<0.01
**Creatinine (mg/dL ± IQR)**	0.35 ± 0.16	0.37 ± 0.16	0.35 ± 0.18		0.89		0.19
**Hospitalization**							
**Yes, n (%)**	99 (11.5)	30 (16.0)	129 (12.3)	1.45 (0.9–2.3)	0.11	1.61 (1.02–2.51)	<0.05
**No, n (%)**	759 (88.5)	158 (84.0)	917 (87.7)				
**All, n (%)**	858 (82.0)	188 (18.0)	1046 (100)				

## Data Availability

The data generated and analyzed during the present study are not publicly available but can be received upon reasonable request to the corresponding author.

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
