# Peer review of "Burden and Clinical Characteristics of Influenza and Its Complications in Children Across Multiple Epidemic Seasons"

_viruses, 2025, doi:10.3390/v17121574_

Round 1

Reviewer 1 Report

Comments and Suggestions for Authors

The manuscript presents a comprehensive retrospective analysis of influenza burden and complications in pediatric patients across three epidemic seasons. The study design is appropriate, data analysis is rigorous, and findings contribute valuable insights to pediatric influenza management. The following revisions are suggested before final acceptance.

  1. Discuss potential selection bias in patient enrollment from the emergency department.
  2. Include multivariate regression analysis to identify independent risk factors for complications.
  3. Present adjusted odds ratios alongside unadjusted values in Tables 2-3.
  4. Compare hospitalization rates and complication patterns with contemporary studies.
  5. Address potential impact of COVID-19 pandemic on influenza epidemiology during study period.
  6. Expand on immunological mechanisms underlying age-related severity differences.

Author Response

Dear Editor-in-chief,

thank you for your revisions and for the very helpful comments on our work.

Please see the attachment with the rebuttal letter.

Sincerely,

Ludovica Trombetta, MD

Maddalena De Peppo Cocco, MD

Reviewer 2 Report

Comments and Suggestions for Authors

The manuscript entitled “Burden and Clinical Characteristics of Influenza and Its Complications in Children Across Multiple Epidemic Seasons” by Dondi A et al., presents a retrospective observational study including 1046 children with  laboratory-confirmed influenza who accessed the Pediatric Emergency Department of a tertiary center in Bologna, Italy, over three consecutive epidemic seasons. The authors investigated the clinical characteristics and severe complications associated with influenza infection requiring hospitalization.

While the study addresses an important topic and includes a large pediatric population, the manuscript would benefit from several improvements, particularly regarding the clarity and completeness of the methodology and the presentation of results.

Major Comments

Abstract

The Abstract should be improved by including quantitative data rather than only descriptive statements. 

Methods

The Methods section would benefit from additional detail, as the current description is quite brief and provides limited information on the study design, patient population, and laboratory procedures.

  • Paragraph 2.1 could be improved for clarity. The manuscript describes a retrospective study including patients who presented to the emergency department and tested positive for influenza A or B. However, it is not explicitly stated whether only a single respiratory sample per patient at diagnosis was considered, or if additional samples collected during hospitalization were included.
  • Paragraph 2.2 could be improved by providing the laboratory methodology, including the type of respiratory samples collected, limits of detection and PCR thresholds for positivity.

Results

  • Regarding pediatric patients with lower respiratory tract infections, could the authors provide more detailed information on the specific clinical manifestations observed (e.g., pneumonia, bronchiolitis)?
  • The authors reported the lack of data on co-infections as a limitation; however, considering that the diagnostic assay allows simultaneous detection of multiple viral respiratory pathogens, a sub-analysis assessing the impact of co-infections on hospitalization or complications would provide additional value to the manuscript.

Discussion

  • It would be useful to include a comment in the discussion about the potential role of viral load in influencing disease severity and the risk of complications, as several studies have suggested an association between higher viral loads and more severe influenza infections.

Minor Comments

Tables

In Table 1, the total number of comorbidities should be checked. In addition, percentages are reported without a decimal digit after the comma. Moreover, the content under the “Season 1” string in the table 2 should be checked.

Table legends and abbreviations

In table 1 and 2 please add a legend clarifying all abbreviations (e.g., IQR, M/F for gender).

Figure

The figure currently labeled as “Figure 2” in the text corresponds to “Figure 1” in the legend. This discrepancy should be corrected, and the panels (currently labeled 2a, 2b, 2c) should be updated accordingly. Additionally, the axis labels and text within the histograms are difficult to read; increasing font size would enhance visibility.

Discussion

To improve readability, it could be helpful to merge the discussion sections into one paragraph.

Author Response

(The authors gave the same response as above.)
